# Implementing a Sustainability Framework in Healthcare: A Three-Lens Framework

**DOI:** 10.3390/healthcare11131867

**Published:** 2023-06-27

**Authors:** Sanjay Rajagopalan, Peter Pronovost, Sadeer Al-Kindi

**Affiliations:** 1Harrington Heart and Vascular Institute, University Hospitals, Cleveland, OH 44106, USA; 2Department of Medicine, Case Western Reserve University School of Medicine, Cleveland, OH 44106, USA; 3Department of Anesthesia and Critical Care Medicine, University Hospitals, Cleveland, OH 44106, USA

**Keywords:** sustainability, environmental, society, governance, pollution, healthcare

## Abstract

The list of sustainability issues that can pose risks to people, society, and healthcare organizations (HCOs) has been steadily growing over the last decade. HCOs and related industries are responsible for greenhouse gas emissions, pollutants, and unsustainable practices with a substantial death and disability footprint. There is an urgent need for health care transformation that advances quality, safety and value in order to address the public health crisis arising from healthcare pollution and to the meet rapidly moving deadlines to avert climate change. Sustainability initiatives are yet further linked with diversity, equity, inclusion, and justice, with HCOs being asked to disclose their commitments to these as part of “good” environmental society and governance (ESG) practices. In this paper, we review approaches to embed sustainability as a core strategy in HCOs and discuss implementation from the standpoint of a three-lens political, strategic, and cultural framework. We discuss solutions to embed sustainability and to facilitate buy-in, and provide a pathway to operationalize sustainability initiatives.

## 1. Introduction

Healthcare organizations (HCOs) and related industries make significant contributions to their communities by providing a wide range of services. They also contribute a significant percentage of pollution, which has an impact on health. Recent estimates have suggested that at least 9 million deaths globally are directly attributable to pollution, a figure that is likely a vast underestimate [1]. HCOs are responsible for as much as 5% of greenhouse gas emissions (GHGs) globally and a considerable proportion of hazardous chemicals and solid wastes that are never recycled. The U.S. healthcare sector alone contributes to roughly 6–8% of all GHG emissions in the U.S. [2,3]. While there is marked regional heterogeneity, per capita emissions for healthcare in the US and Europe are among the highest in the world. The US accounts for 25% of global health sector emissions, the highest proportion attributable to any country [4]. Operational emissions and heating and cooling (Scope 1 and Scope 2) make up 30% of US emissions, while Scope 3 emissions (supply chain) comprise 70% [5]. Figure 1 illustrates the three healthcare emission scopes. This estimate is likely true for the rest of the world, where Scope 3 emissions attributable to HCOs dominate compared to Scope 1 and 2, given the outsized reach of HCOs on the economy at large (>20% of the GDP for the US). Despite the considerable pollutant and carbon footprint of the healthcare industry, the sustainability reporting, pollutant, and emission generation from HCOs are not mandated nor commonly monitored by executives and boards. HCOs also contribute substantially to waste, including medical waste and plastics [6]. Many procedures and in-hospital care including surgery, have significant environmental impacts, including CO2 emissions that contribute to climate-related threats to human health. Importantly, many opportunities to effectively reduce environmental impacts exist and can provide economic, health, and social co-benefits [7,8]. An integral component of sustainability is the ability to embrace diverse communities and the unique voices they support [9]. Indeed, the greatest impact of climate change is often on under-resourced communities or in parts of the global south, which tragically, have not contributed to the climate crisis but are nevertheless on the frontlines of bearing the impact of its consequences.

## 2. Sustainability in Health Care

The burden of sustainability issues that can pose risks to people, society, and HCOs has been steadily increasing over the last decade. An integrated discipline of “sustainability” is rapidly evolving, which includes contributions from the fields of industrial and environmental engineering, sustainability science, medicine, public health, economics, and management [10]. While an effective sustainability agenda is seen as vitally important for the corporate sector, not only as a competitive strategy but also as a key service to the community and the planet, this does not seem to have been adopted by many HCOs [11]. One reason for this could relate to the primary mission of healthcare, which is to save lives, with sustainability not imbuing as much of a sense of urgency, which is almost always the case when dealing directly with healthcare issues. An unintended consequence of this lack of exigency, is that healthcare facilities currently do not see the need to monitor and report on their pollution or consumption the way that corporations and other institutions are often pressed to do [12]. Yet, there could not be a more important time for HCOs to take a concerted stance and make sustainability part of their strategic agenda. Unlike a “for-profit corporation”, there are no investors or key levers that would drive the measurement and disclosure of these data. This could, however, be changing soon, with governmental, other public health, and financial agencies starting to request that HCOs disclose their carbon footprint as part of “good” environmental society and governance (ESG) practices [13]. Additionally, patients and stakeholders in healthcare remain attached to misconceptions around what sustainability in healthcare means. For instance, to some, sustainability is about financial rectitude alone, to others, it is about recycling, but for most, the scope and scale of the effort are not completely comprehended. Compounding these misconceptions, the C-suites and boards of most HCOs have little representation or expertise in environmental and sustainability domains. The introduction of sustainability “think” in HCOs will thus take substantive efforts and will challenge most organizations, but it is important for a variety of obvious reasons [14].

In this paper, the rationale and implementation strategies for HCOs that wish to develop a robust sustainability agenda are provided. Examining a sustainability agenda from the vantage point of the strategic, geopolitical, and cultural domains of HCOs, sometimes referred to as the “3-lens organizational framework”, allows one to not only appreciate barriers that could impede progress but also shed light on pathways for successful transformation [15]. This framework includes three aspects: strategic (which includes processes and procedures), political (authority and power), and cultural (underlying attitudes and beliefs). Envisioning sustainability in HCOs may require the use of these three domains to visualize challenges and opportunities to implement sustainability planning for HCOs.

HCOs can then easily lead the way in sustainability initiatives and play their part in addressing climate change and environmental pollution, all while making good on their financial targets and commitments to their stakeholders and society at large.

## 3. Sustainability as a Bedrock of HCO Business Strategy

HCOs must start to consider their environmental impact and how they can shift their practices to a more sustainable framework to improve the health of the communities they serve [2,3]. Sustainability initiatives are closely linked with diversity, equity, inclusion, and justice [16]. Historically, those most affected by pollution, climate change, and environmental degradation are minorities and other susceptible individuals living in vulnerable communities [17]. Investment by HCOs in the communities they serve is currently minimal beyond reimbursed healthcare delivery, with almost no investments in community sustainability.

For example, while many HCOs excel at delivering excellent healthcare services, they often lag in providing community services, such as strategies to improve the environment (e.g., tree plantation, green space development), community-based screening efforts, and investments in healthy neighborhoods (e.g., smart city design). While community outreach and engagement are professed by many nonprofit organizations, investments in sustainable living, structural improvements in the physical environment to reduce environmental pollution exposures, and resiliency measures to combat climate change in communities that HCOs serve are minimal to non-existent.

Most HCOs have little experience building sustainability initiatives as part of a comprehensive ESG strategy, with no expertise to navigate a complex and evolving field. Additionally, sustainability initiatives require changes in infrastructure, long-term institutional commitments that are well beyond the average half-life of a Chief Executive Officer, with downstream socioeconomic adjustments that may not be immediately apparent. For example, sufficient capital investments over a longer-term horizon may be required to transition to low-emissions assets, responsibly retire high-emissions assets, secure energy needs through long-term contracts from renewable sources, etc. The current fragile state of HCOs has essentially stripped most organizations of the ability to make any investments towards sustainability. However, if HCOs do not manage the cost impact of decarbonization, and communities that they serve are left behind and not included in sustainability initiatives, then the transition to a net-zero economy at large may be put at risk, given the magnitude to which HCOs are wedded to other sectors in the economy. Figure 2 shows the strategic and moral imperatives for HCOs to embrace sustainability.

## 4. Mapping the Political Terrain and Facilitating Political Buy-In

To overcome the profound innate resistance for HCOs to change their sustainability practices and address these challenges proactively, there is a need to chart the political terrain viewed through the lens of four metaphoric domains: the weeds, the rocks, the high ground, and the woods (Figure 3) [18]. This mapping exercise could help identify key barriers, both implicit and explicit, and their respective contexts. Importantly, such an exercise is critical in designing interventions that can serve to eliminate these barriers.

A commitment chart, as well as stakeholder mapping techniques, may be used to map out key members and/or units that could serve as allies/testbeds or, conversely, as bottlenecks [19]. Once stakeholders are mapped and the alignment of members is accomplished, additional techniques for buy-in may be required. This includes securing public commitments from the Chief Executive Officer (CEO), the Chief Operating Officer (COO), and the Board on a new focus on sustainability and environmentally sound practices. This includes getting members in the C-suite to commit resources and using “escalation of commitment” and “perceptions of influence” approaches [15].

Both formal and informal networks within an organization that currently sustain poor environmental practices would need to be identified and realigned. Some examples of formal mechanisms include reviewing current contracts with energy suppliers, and evaluating transportation fleets and supply chain entities. Medication and hospital equipment shortages may interfere with procurement, disposal of waste, water management, and any other entities that may fit with a sustainability agenda are also fair targets. Another example is the choice of foods at the hospital and the adherence to recycling standards. A concerted effort to reduce plastic consumption and disposable items, rethinking recycling programs, could pay off quite rapidly, helping to reduce costs and improve overall hospital bottom lines. Alignment of hospital food practices that are healthy and environmentally sound (predominantly plant-based and locally sourced with an emphasis on fresh foods) will require the disruption of existing formal and informal relationships with current supply chains, cafeteria staff, and creation of newer relationships.

Governmental incentives to decarbonize could accelerate HCO transitions. These could include financial incentives for decarbonization, integrating HCO emissions metrics with incentives to lower emissions, improve health outcomes, and enhance value in a “value-based” healthcare framework.

## 5. Mapping Strategy and Enabling Strategic Transformation

Creating and maintaining a sustainability framework begins by visualizing and analyzing the status of sustainability in the organization and articulating a new bold vision for the future, using a framework that can be understood. For results-oriented organizations that value revenue targets and quality outcomes, the “why we do what we do” is as important as “what we do when we do”. Figure 4 details seven high-impact actions from the Global Road Map for Health Care Decarbonization that may be worth incorporating into HCO mission statements. Below, we articulate three different strategic goals that may be considered.
Embed sustainability in the organization’s strategy and purpose. HCOs have the opportunity to move beyond platitudes and build action-oriented sustainability efforts throughout their business, from supply chains to the overall environmental footprint. These steps should align with the organization’s overall mission and vision. HCOs should establish overarching intentions by including sustainability principles in their mission statements. They can also signal their commitments through explicit commitments towards climate change coalitions and establishing precedents by demanding adherence to “sustainable” codes of conduct from supply chain partners. Such declarations are intentional and powerful. By publicizing intent and disseminating public awareness about sustainability and corporate responsibility, HCOs send a powerful message, given the alignment of improved health outcomes with sustainability measures. This could range from reducing waste, eating plant-based locally sourced diets, and reducing transportation and other energy-intensive activities [20,21,22]. Declaration of a clear-cut sustainability agenda may also allow organizations to differentiate themselves from their competitors and enhance reputations with customers, employees, investors, and analysts.Sustainability metrics for transparency and accountability. HCOs may be able to make significant progress in their sustainability efforts by better measuring, quantifying, communicating, and sharing the value they create for society. HCOs would need to measure their GHGs and map out Scope 1, 2, and 3 emissions, and put in place approaches to track emissions, especially across supply chains that are integral for understanding Scope 3 emissions [3,23]. HCOs can then use these assessments to prioritize and plan net-zero strategies, reducing waste, fixed costs, and finally securing plans to adapt, decarbonize, and thrive in a net-zero economy. HCOs can take a proactive approach to telling their story to employees, customers, suppliers, and other stakeholders using these measures. However, one must acknowledge that current metrics to measure sustainability, especially in the environmental domain, although the easiest to track, are often vague. Newer approaches in measuring emissions, including the use of blockchain technologies and cost accounting approaches, have been advocated and may result in better metrics and disclosures [24].Create practical and transparent goals. Creating value and impact through sustainability means viewing it as more than a charity effort and developing a practical plan that can be acted on. Figure 3 provides a list of 7 high-impact actions that represent clear actionable goals. Some of these, such as Item 5, entail engagement with suppliers in order to reduce Scope 3 emissions and may involve a much longer time horizon. The lowest hanging fruit within the workplace may include rethinking the cafeteria menu, well-being services, gym memberships, and better work environments, for example, lighting, ergonomic desks, and indoor air quality. More recently, it has emerged that mental health is at least as important as physical health, and it’s likely that early intervention and prevention of physical and mental health issues will be more effective than trying to control absenteeism and healthcare costs. The health impact of the emissions of HCO may need active attention. For instance, the emissions intensity of an HCO and the energy intensity may negatively impact its populations through continuing health impact, with costs that are borne by the communities the HCO serves and/or by taxpayers (a negative externality in economic terms) [1,17]. For companies managing their output within an ESG framework, now can proactively engage in improving public health by self-regulating before regulators impose fines, product bans or punitive taxation. Taking proactive steps, such as rethinking material-sourcing and organizational processes to promote rather than impair health, are not only commendable but also good foundational business principles.

## 6. Mapping Culture and Accomplishing Cultural Transformation

Leaders must understand how the current culture in a non-sustainable healthcare framework precludes the attainment of current goals, or conversely, how sustainability initiatives can facilitate better margins and health outcomes. In healthcare, given the rapidly changing nature of the industry, a “learning environment” (while maintaining a focus on results) may be appropriate. Unlike strategy and politics, culture is context-specific, and changing culture must involve an intimate understanding of the emotional and social dynamics of people in the organization [25]. Four practices, or levers, can help in cultural transformation.
Inside-out Listening: This approach helps provide insights into what defines the culture of a company through conversations with board members, the C-suite, star employees, and discussing what sustainability means to the HCO and how it aligns with the culture of the organization. Sustainability issues are essentially linked to the social mission of most HCOs, and this linkage can often find renewed enthusiasm and rejuvenation of purpose and mission. At the same time, attention may need to be paid to the extrinsic demands of stakeholders and overriding concerns such as the pandemic, which may essentially drown out any conversations on sustainability.Visualizing Culture and Change Champions: Much like implementing a new strategy, creating a new culture should begin with an analysis of the current culture. Most HCOs operate in a result- and authority-oriented culture, where embracing an uncertain and evolving world of sustainability and shifting toward learning or enjoyment (while maintaining a focus on results) may be appropriate. Because of culture’s ambiguous and hidden nature, helping people better understand and connect to the need for change is critical in cultural transformation. Leaders and their designated surrogates serve as important catalysts for change by encouraging and enforcing sustainability practices. Those who are unsupportive of the desired change can be engaged and re-energized through training and education about the important relationship between sustainability and strategic direction. These same individuals could be asked to lead demonstration projects in sustainability or act as change champions.Conversations for Culture Change: Conversations such as road shows, listening tours, and structured group discussions can support cultural change. Social media platforms may help encourage conversations between managers and employees. Sustainability champions can advocate for a culture shift through their language and actions. As employees start to recognize that their leaders are talking about new business outcomes—such as innovation instead of quarterly earnings, for example—they will begin to behave differently themselves, creating a positive feedback loop.Culture Shift and Reinforcement through Organizational Design: When a company’s structure and processes are aligned to support the vision and strategy, transforming culture and behaviors can be less onerous. Organizational design features can have an outsized impact on how people think and behave. For example, sustainable practices in the cafeteria, with significant attention paid to signage and explicit and subliminal messaging on sustainability (such as messaging on screen savers, walls, and educational materials), may help reinforce sustainable behaviors in other parts of the organization. Training practices can further reinforce the target culture through required annual certification in sustainability practices.

## 7. Operationalizing a Sustainability Initiative

A strong operational plan for engaging, standardizing, and maintaining sustainability initiatives is essential. We outline key steps HCOs can take on their sustainability journey.
Choosing a Pilot Indicative Unit for Sustainability: Most HCOs standardize processes by testing them on a small scale in one pilot unit, which serves as the “model unit” or a place for experimentation and implementation. When looking for a model unit, some attributes are particularly important. (a) Stability and Opportunistic Units: the best units to test sustainability initiatives may include those with low workforce turnover and ample sustainable practice opportunities, such as operating rooms. Alignment around sustainability goals: managers in the unit should understand what is expected of them, how their work is going to change, and why standardizing practices is important for sustaining quality. (b) Practice “hygiene”: it is preferable to find testbeds that already have a few sustainable practices in place and stable routines. A chaotic environment will have difficulty implementing sustainable practices. (c) Engagement: the unit should have a respected local change champion who can build excitement for change, encourage participation, coach the team, and celebrate success.Deploy Overlapping Timelines: Overlapping timeframes for accomplishing goals are a way to address complex and ambiguous challenges such as sustainability. Thus, having short-term, medium-term, and long-term goals can make progress seem more tangible. Short-term goals (months to a year) could include the reduction of plastics and waste, aggressive recycling, the shift to clean energy sources, energy conservation approaches, and cafeteria reform to include locally sourced foods and plant-based diets. In the medium term (years), priorities may include a total switch to electric cars, engagement with suppliers to design a more circular system, and reducing Scope 3 emissions. Ultimately, the long-term target could be achieving net carbon neutrality across the HCO.Process Evaluation of Sustainability Initiatives: A life cycle analytical (LCA) approach can help account for inputs, emissions, and subsequent health impacts from “cradle-to-grave”. This includes direct emissions from product use, along with indirect emissions from both upstream (i.e., the supply chain production and transportation) and downstream (i.e., waste disposal management) activities [26]. LCA and other industrial ecology methods and tools, such as circular economy principles, can be used to assess clinical pathways, procedures, individual drugs, and medical devices. Assessing the adoption, implementation, and maintenance of initiatives using an implementation science framework is crucial to ensure the durability of change [27]. Additionally, evaluating the cost-effectiveness of sustainability initiatives is important to provide reassurance that these measures improve the financial bottom line of cash-strapped HCOs.

## 8. Conclusions

Currently, healthcare pollution causes significant direct and indirect public health damage, resulting in substantial costs and resource consumption. Sustainability initiatives will require a transformational shift to new models of care that are crucial to achieving net zero status, as outlined by the Intergovernmental Panel on Climate Change (IPCC), within a short timeframe. Healthcare has a moral and economic imperative to accelerate sustainability efforts. The time to start is now.

## Figures and Tables

**Figure 1 healthcare-11-01867-f001:**
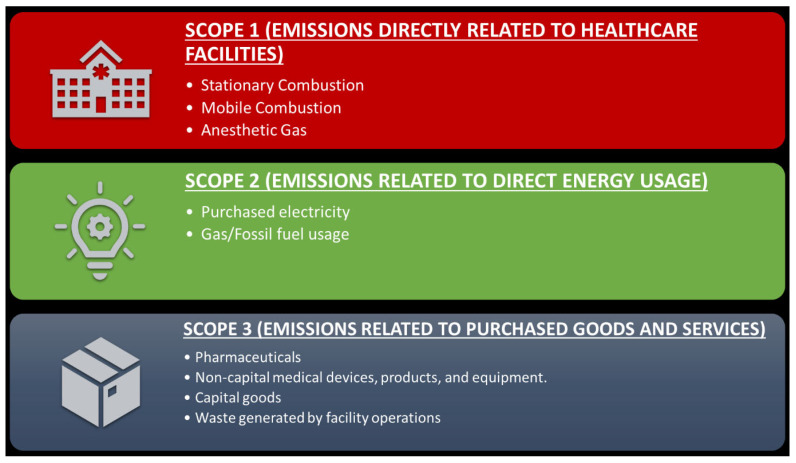
Greenhouse gas emissions from healthcare organizations.

**Figure 2 healthcare-11-01867-f002:**
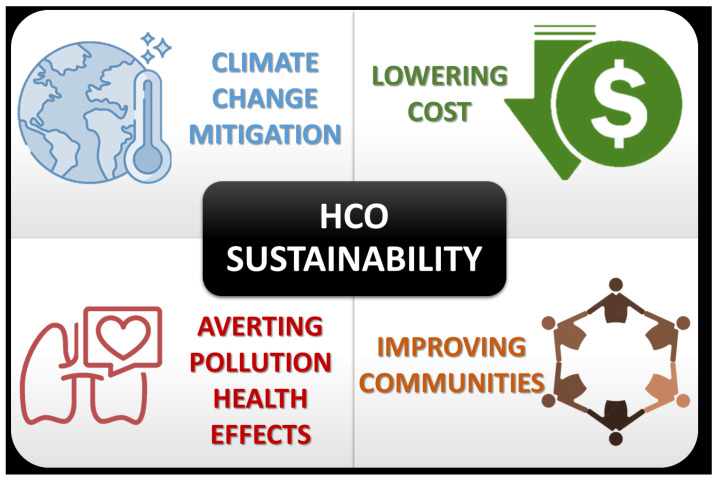
Strategic and moral imperatives for healthcare organizations to embrace sustainability.

**Figure 3 healthcare-11-01867-f003:**
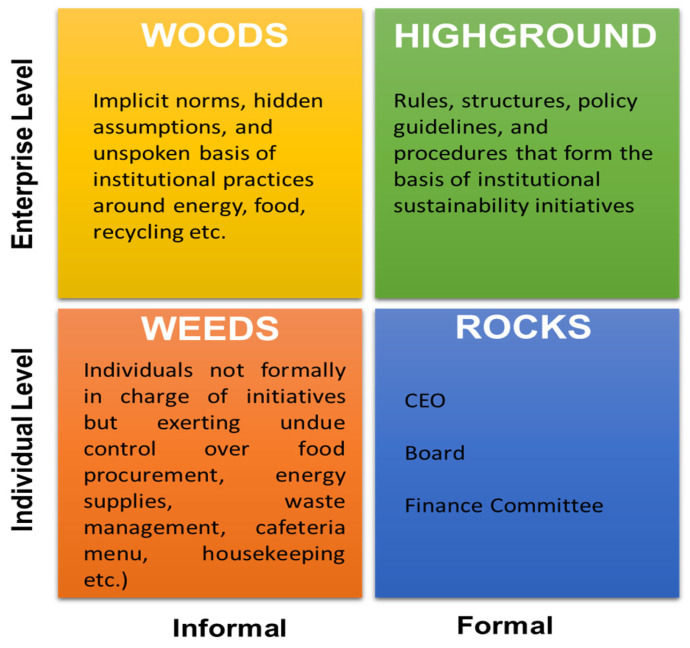
Organizational political considerations. Woods, high ground, weeds and rocks. (Adapted from: The 4 Types of Organizational Politics by Michael Jarrett. Harvard Business Review 2017) [16].

**Figure 4 healthcare-11-01867-f004:**
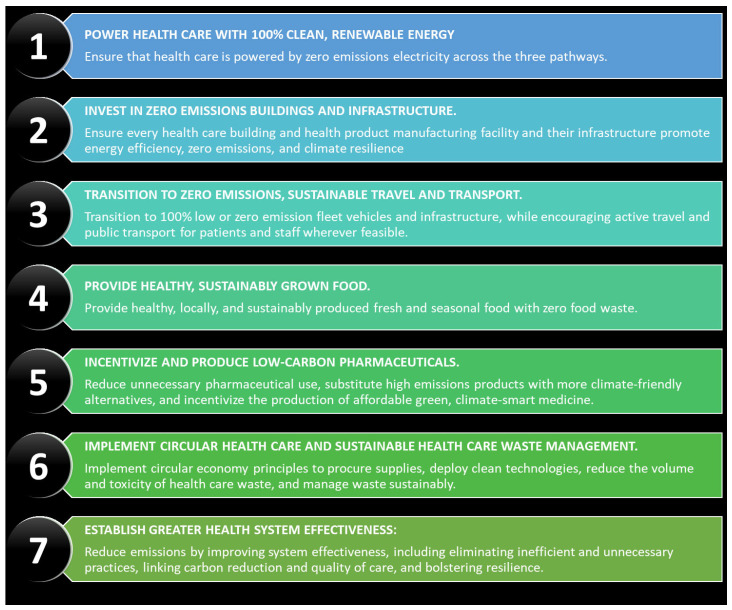
Seven high-impact actions from the Global Road Map for Health Care Decarbonization. From, Global Road Map for Health Care Decarbonization. A navigational tool for achieving zero emissions with climate resilience and health equity. Health Care Without Harm Climate-Smart Health Care Series. Executive Summary. Green Paper Number 2. Available online: https://healthcareclimateaction.org/roadmap (accessed on 14 May 2023).

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
