# Peer review of "Implementing a Sustainability Framework in Healthcare: A Three-Lens Framework"

_healthcare, 2023, doi:10.3390/healthcare11131867_

Round 1

Reviewer 1 Report (Previous Reviewer 2)

Although improvements were made, I noticed some issues that still need to be corrected:

·        The designation ”Figure” was added to: “…Create practical and transparent goals. Creating value and impact through sustainability means viewing it as more than a charity effort and developing a practical plan that can be acted on. Figure 3 provides a list of 7 high impact actions that represent clear actionable goals…”.  This is not the content of Figure 3.

· On page 1, under “Operational emissions and heating and cooling (Scope 1 and Scope 2) make up 30% of U.S emissions, Scope 3 emissions (supply chain) contributes to 70% (Figure 1)”, the US is mentioned but the figure is generalist, without any geography mentioned. This figure should be improved to clarify this issue (for instance its legend). Is it global or just the US?

·        I noticed that Figures 1 and 2 remain without source…but maybe these were produced by the authors themselves? If that is the case, please ignore this comment.

·        The type of paper mentioned in the platform is currently Opinion, but in the pdf, it remains as a Review. This should be corrected to match, since in my view, the appropriate type is Opinion.       

I noticed that some newly written phrases in English are poorly written/constructed, and therefore the text should be revised, to provide a clear and accurate message to readers. 

Author Response

Although improvements were made, I noticed some issues that still need to be corrected:

  • The designation ”Figure” was added to: “…Create practical and transparent goals. Creating value and impact through sustainability means viewing it as more than a charity effort and developing a practical plan that can be acted on. Figure 3 provides a list of 7 high impact actions that represent clear actionable goals…”.  This is not the content of Figure 3.

RESPONSE: Thank you. Figure 3 in the revised version has the four components (woods, highground, weeds, and rocks).

  • On page 1, under “Operational emissions and heating and cooling (Scope 1 and Scope 2) make up 30% of U.S emissions, Scope 3 emissions (supply chain) contributes to 70% (Figure 1)”, the US is mentioned but the figure is generalist, without any geography mentioned. This figure should be improved to clarify this issue (for instance its legend). Is it global or just the US?

RESPONSE: this is now corrected. “Operational emissions and heating and cooling (Scope 1 and Scope 2) make up 30% of U.S emissions, Scope 3 emissions (supply chain) contributes to 70%. Figure 1 illustrates the 3 healthcare emission scopes.6

  • I noticed that Figures 1 and 2 remain without source…but maybe these were produced by the authors themselves? If that is the case, please ignore this comment.

RESPONSE: Thank you. Yes the figures were generated by us.

  • The type of paper mentioned in the platform is currently Opinion, but in the pdf, it remains as a Review. This should be corrected to match, since in my view, the appropriate type is Opinion.   

RESPONSE: You are correct. The paper is an opinion.    

  • I noticed that some newly written phrases in English are poorly written/constructed, and therefore the text should be revised, to provide a clear and accurate message to readers. 

RESPONSE: Thank you. We revised the language throughout

Reviewer 2 Report (Previous Reviewer 3)

Implementing A Sustainability Framework in Healthcare: A Three-Lens Framework

Reviewer Report Round 2

·         Even if the abbreviation is mentioned in the abstract, the abbreviation should be used after it is explained in the first place in the text. For example; "HCOs"

·         Some of the headings were capitalized. Why was it written like that? How was the transition between titles made? It should be prepared more systematically.

·         This should be explained systematically, as the study was not specified as a material method section.

·         Figures should be given in order following the place where they are specified.

·         It is still unclear what the type of study was. Review, opinion etc.? For this reason, it is difficult to evaluate the study and it still does not provide the systematic that should be in any scientific study. It is noteworthy that the study was conducted in the design of a book chapter, not for publication in a scientific journal, and its narration is like a column.

·         The systematic of the study should be clear and understandable on the headings. There should be a meaningful transition between headings.

·         There are still lacks in the preparation of the manuscript according to the journal template.

·         And finally, the responses to the initial evaluations are lack and not satisfactory. And there are studies in the literature that can be associated with this study.

A moderate editing of English language is still required.

Author Response

  • Even if the abbreviation is mentioned in the abstract, the abbreviation should be used after it is explained in the first place in the text. For example; "HCOs"

RESPONSE: Thank you. This was added

  • Some of the headings were capitalized. Why was it written like that? How was the transition between titles made? It should be prepared more systematically.

RESPONSE: Thank you. This was adjusted.

  • This should be explained systematically, as the study was not specified as a material method section.

RESPONSE: Thank you. As noted above, this is an ‘opinion’ paper

  • Figures should be given in order following the place where they are specified.

RESPONSE: Thank you. The references to the figures are now sequential.

  • It is still unclear what the type of study was. Review, opinion etc.? For this reason, it is difficult to evaluate the study and it still does not provide the systematic that should be in any scientific study. It is noteworthy that the study was conducted in the design of a book chapter, not for publication in a scientific journal, and its narration is like a column.

RESPONSE: Thank you. This is an ‘opinion’ paper.

  • The systematic of the study should be clear and understandable on the headings. There should be a meaningful transition between headings.
  • There are still lacks in the preparation of the manuscript according to the journal template.

RESPONSE: Thank you. The editorial staff has now adapted the manuscript into the journal template.

  • And finally, the responses to the initial evaluations are lack and not satisfactory. And there are studies in the literature that can be associated with this study.

RESPONSE: Thank you. This is a short ‘opinion’ piece, and thus we have cited the major studies.

  • A moderate editing of English language is still required.

RESPONSE: Thank you. We revised the language.

Reviewer 3 Report (Previous Reviewer 4)

The manuscript can be accepted. 

However, there are a few typographical errors that need to be addressed.

Also, the references should be properly formatted as per the journal's guidelines (including DOIs). 

The English is acceptable.

Author Response

The manuscript can be accepted. 

  • However, there are a few typographical errors that need to be addressed.
  • Also, the references should be properly formatted as per the journal's guidelines (including DOIs). 

RESPONSE: Thank you. We have adapted the endnote style provided by the journal under instruction to the authors section (which does not seem to include DOI). We would be happy to provide DOIs if needed by the editorial team.

Round 2

Reviewer 2 Report (Previous Reviewer 3)

The authors have satisfactorily revised what I have pointed out. I think the study is ready for publication.

Minor editing of English language required.

This manuscript is a resubmission of an earlier submission. The following is a list of the peer review reports and author responses from that submission.

Round 1

Reviewer 1 Report

A sustainability framework in healthcare refers to a set of principles, strategies, and practices designed to promote long-term ecological, social, and economic viability in the delivery and management of healthcare services. It encompasses a wide range of areas, such as energy conservation, waste reduction, responsible procurement, community engagement, and patient safety. The framework aims to improve the quality of care while minimizing the negative impacts of healthcare on the environment, public health, and resources. By adopting a sustainability framework, healthcare organizations can reduce costs, increase efficiency, enhance their reputation, and contribute to the well-being of patients and the wider community. This approach recognizes the interconnectedness of health, environment, and society and seeks to create a sustainable future for all. The manuscript entitled “Implementing A Sustainability Framework in Healthcare: A Three-Lens Framework” has covered all these issues in detail. Implementation of a sustainability framework in healthcare has been discussed in a comprehensive manner in this article by explaining, a three-lens framework, which can be used to guide healthcare organizations through the process. The first lens involves adopting a comprehensive approach that incorporates environmental, social, and economic factors. This involves understanding the full impact of healthcare operations on the environment, public health, and resources, and identifying opportunities for improvement. The second lens involves engaging stakeholders, including patients, employees, suppliers, and the wider community. This involves building partnerships and collaborations, promoting transparency and accountability, and empowering stakeholders to drive change. The third lens involves embedding sustainability into organizational culture and practices. This involves setting clear goals and targets, measuring and monitoring progress, and providing education and training to staff. By applying this three-lens framework, healthcare organizations can create a sustainable future for all and improve the quality of care for patients. The manuscript entitled “Implementing A Sustainability Framework in Healthcare: A Three-Lens Framework” has covered all these issues in detail. The article is very well written, easy to understand and comprehensive in nature.

Best wishes for future endeavors.

Reviewer 2 Report

Although this article seems well written in terms of English, and its message seems clear (in the sense that it is easy to read and understand), I suggest below a number of critical issues that need to be addressed, in my view:

According to the definition of a review article on the instructions for authors: “Reviews offer a comprehensive analysis of the existing literature within a field of study, identifying current gaps or problems.”

I do not detect a comprehensive analysis of the existing literature, since there are only 12 literature references cited, and no review methodology is declared, therefore it is difficult to assess if the definition/objective of a review is achieved.

Also, many statements along the text do not seem to be supported in the literature. Just to provide some examples, on page 2: “Additionally, patients and stakeholders in healthcare remain attached to misconceptions around what sustainability in healthcare means.” Which misconceptions? And where is this supported in the literature?    

Also on page 2: “Examining a sustainability agenda from the vantage point of strategic, geopolitical and cultural domains of HCOs (sometimes referred to as the “3 lens organizational framework”)”. Sometimes referred by which authors?

Related to this, and also according to the definition, “The structure can include an Abstract, Keywords, Introduction, Relevant Sections, Discussion, Conclusions, and Future Directions…”

The fact that there seem to be no formal chapters dedicated to discussion and conclusions, coupled with the absence of comprehensive literature support and a review methodology, makes it hard to assess which statements, proposals and critical analysis originated in the authors, and which came from previous literature.

The figures do not have a source, so I suggest it should be included.

There is a support element mentioned in the text that does not seem to be present. In the last line of page 4, there is a (figure, table…?) “3” that “provides a list of 7 high impact actions…”, but I cannot detect where this list is.  

Reviewer 3 Report

Implementing A Sustainability Framework in Healthcare: A Three Lens Framework

Reviewer Report

I read and reviewed the study with great interest. I thank the authors for their study, which takes an adequate approach to discussing how society, health, and the environment interact from a strategic, political, and cultural perspective. I think it will be ready for publication after the revisions below:

-   Please, merge the first 3rd and 4th lines of the introduction.

- Many noteworthy sentences have not been cited throughout the manuscript. Please cite with current articles.

- Please cite the sentences between the 2nd and 3rd citations with current literature.

- Please re-arrange the main headings and subheadings to the same standard.

-The material and method section, which includes the study’s design, data collection, and evaluation criteria, should be added.

-Adding the Results and Discussion section in which the current findings are compared to the literature will make the subject more understandable and clear.

-Since the subject has been handled from a very comprehensive perspective, a Conclusion section should be added and should present a final awareness, suggestion and/or conclusion for the readers.

- The study should be improved by the current literature and the references should be increased a little more. Also, references without volume, issue and/or page numbers should be revised.

- Please revise your manuscript in accordance with the journal template.

Reviewer 4 Report

In this manuscript, the authors have provided an overall review and further approaches to embed sustainability as a core strategy in HCOs and their implementation from the standpoint of a “3-lens” political, strategic, and cultural framework. Although the manuscript provides an overview of implementing a sustainability framework, the manuscript lacks to emphasize the major issue of carbon footprints and waste disposal. The manuscript needs to be polished in terms of the contents of the organization of the manuscripts, proper examples with existing methodologies in the literature, and their shortcomings. Providing real-time examples where the sustainability framework has already been established to a certain level can be correlated. The manuscript cannot be considered for publication in its current stage.

11. How much pollutants do HCOs contribute to overall emissions? What factors contribute to the majority of carbon footprints in HCOs?  

22. References should be properly cited.

33. The manuscript needs to be extremely refined in terms of framing, the right choice of words, and also the contents of each paragraph for ease of understanding. Particularly some abrupt points can be seen, which break the flow, thus making it difficult to correlate and understand.

44. Much of the paragraph provided in “Political Terrain and Facilitating Political Buy-in” is focused on examples of food-related sustainability. More appropriate examples in concern to the title of the paragraph must be included and correlated to understand them more clearly.

55. The Figures are not clearly discussed enough. The paragraphs can be framed around the figures to bring out the concepts more clearly to increase relevance and understanding.

66. In “Embed Sustainability in the organization’s strategy and purpose”, the sentence “reducing pollution emissions, eating plant-based locally sourced diets and reducing transportation and other energy-intensive activities” can be elaborated and provided with proper references.

77. On page 4, last line, the word is missing before” ……. provides a list of 7 high impact actions that represent clear actionable goals”

88. As stated in the abstract, “Sustainability initiatives are further heavily linked with diversity, equity, inclusion, and justice”, how are these linked? Providing proper examples, these points can be highlighted in the manuscript.

99. A brief summary can be included at the end of the manuscript.